# Comparison of the efficiency of Deebag and jute made bag for faecal sludge management and wastewater treatment

**Mehedi Hasan Mishuk**[1]*, **S. M. Tariqul Islam**[2], **Muhammed Alamgir**[1]

**1** Department of Civil Engineering, Khulna University of Engineering & Technology, Khulna, Bangladesh,
**2** Institute of Disaster Management, Khulna University of Engineering & Technology, Khulna, Bangladesh

* mehedihasan@kuet.ac.bd

**Data Availability Statement:** All relevant data are within the paper and its Supporting Information files.

## Abstract

Faecal Sludge Management (FSM) has become a prominent environmental concern in the today's world. Dewatering of sludge and the treatment of wastewater (WW) are the prime spiny issue because of the deleterious essence of faecal sludge (FS) and WW in the environment. The main focus of this study was on FSM by 'Deebag' and 'Jute Bag' through dewatering and filtering. Deebag is a dewatering as well as filtering media which is made with geotextile and polypropylene. Contrariwise, three types of jute bags were made of jute fiber for using as the same purposes of Deebag. A polyacrylamide polymer was used in this study and both filtering and dewatering were done in two ways—with and without the presence of polymer. Biochemical Oxygen Demand at 5 days ($BOD_5$), Chemical Oxygen Demand (COD), pH, Electrical Conductivity (EC), Chloride ($Cl^-$), Phosphate ($PO_4^{3-}$), Nitrate ($NO_3^-$), Total Suspended Solid (TSS) and Total Dissolved Solid (TDS) parameters of raw and filtering samples were analyzed to assess the performance of WW treatment by Deebag and jute bags. Only using polymer was observed as one kind of treatment of WW. Deebag has been found to show the maximum dewatering capacity as well as treatment efficiency comparing with the jute bags. However, among three types of jute bags, double jute layered bag has shown the best performance. Maximum dewatering for Deebag and jute bags were found 88% and 83% respectively while using the polymer.

## Introduction

Management of faecal sludge (FS) in an appropriate and sustainable approach is now a major concern in the world. Because, Lack of proper management of FS has become a source of the transmission of many infectious diseases, including cholera, typhoid, hepatitis, polio, cryptosporidiosis, ascariasis, and schistosomiasis [1]. With the rapid growth of population in the world, biological sewage sludge is being generated at a higher rate that of course should be disposed properly [2]. Solutions for effective and sustainable faecal sludge management (FSM) present a significant Global need [3]. In many developed countries, FS with WW is treated with sewage treatment plant, whereas this scenario is not very common in the developing

**Funding:** The authors received no specific funding for this work.

**Competing interests:** The authors have declared that no competing interests exist.

countries. Without proper management, FS is often allowed to accumulate in poorly designed pits, and WW is discharged into storm drains and open water, or is dumped into waterways, wasteland, and unsanitary dumping sites [4]. The World Health Organization (WHO) estimates that 2.2 million people die annually from diarrheal diseases and that 10% of the population of the developing world are severely infected with intestinal worms related to improper waste and excreta management [5,6]. It is evident that the management of FS is a critical need that must be addressed, and that it will continue to play an essential role in the management of the global sanitation into the future [3].

FSM is also a vital concern in Bangladesh. In Dhaka city, only 20% of the population are under the coverage of highly expensive sewerage network and the rest of the population are using on-site sanitation system such as septic tanks, pit latrines, unhygienic latrines or none at all [7]. The proper management and appropriate treatment of FS in Bangladesh is only a small percentage and no environmental friendly or environmentally sound system for collection and disposal of FS in Bangladesh is present now [8]. The outlet of most of the septic tank is connected with public sewer. Consequently, untreated WW is directly discharged into the public drain. Moreover, there is no proper emptying mechanism for pits or septic tanks. In most cases, it is done manually by sweepers when the problem becomes visible by overflowing or creating a nuisance. Emptying through using the mechanical suction device, known as vacutug is very limited [8].

The third largest city of Bangladesh is Khulna, which is situated in the southwestern part of the country lying in the delta of the river Ganges [9]. Around 1.6 million population has lived in this city with 66,257 number of households [10]. Unfortunate that, Khulna is in lack of sewerage system with its big population [11]. Recently, a designated site has been developed in Rajbandh near Khulna city for dumping FS safely or the treating FS and WW [10]. Some vacutugs are available to carry FS to the designated site under the operation and maintenance of Khulna City Corporation (KCC). Vacutug is a truck which has a mounted cylinder container laying down on the back site of the truck to carry FS and WW from collection site to the diposal site or treatment plant. Due to lack of available spaces in the city area, city dwellers have to reuse their on-site sanitation system. However, emptying of on-site sanitation system is still in a big problem in Khulna city. Though some vacutugs are available, in most of the cases, WW is discharged into the public drains by the city dwellers due to carrying cost of vacutug. Discharged WW is finally gone to either ponds or river. Many people have been suffering from infectious diseases by using those pond or river water. Discharging this polluted water to the environment has significant negative impacts [3].

Dewatering is one of the most important treatment mechanism and it is necessary prior to resource recovery for applications such as composting, or combustion as a fuel [3]. Because in FS, water is found either in bound forms or in free and the free water is fairly easy to remove from the FS, while removal of the bound water is much more difficult [12]. Moreover, removing of microorganisms from water is difficult while they are physically bound to solids and it needs the addition of chemicals or the use of centrifugation, pressure or evaporation [3]. Besides, FS composes of 75% water in an average and this water is heavy and expensive to transport to the disposal or treatment site [13].

This study was aimed at FSM by dewatering and filtering treatment of WW through Deebag and three types of jute bags. Treatment efficiency of Deebag and jute bags was also focused in the study. Deebag is a new concept manufactured in Malaysia for dewatering the faecal sludge with filtration capacity. But Deebag is costly and not available in Bangladesh. Considering these issues, three jute bags were made from locally available burlap and used for dewatering of FS with filtering and comparing the performance with Deebag. Dewatering capacity and WW treatment efficiency of both Deebag and jute made bags are also evaluated in the study.

## Materials

### Study area

The study area is Khulna University of Engineering and Technology (KUET) which is situated in the north-west border of Khulna city and graphically lies at 22"50' north latitude and 89"50' east longitudes [14]. The study area is almost the far most point from the Rajbandh FS disposal and treatment site. So, it was hypothesized that more trips by vacutug may be required for disposal of FS from KUET campus to the designated site of KCC. Naturally, it would cost more than any other places of the city for safe disposal of FS and that might have potentially encouraged the people to empty their FS storage pit by other ways.

### Collection and preparation of Deebag

Tentate Geosynthatics Asia Sdn. Bhd., a Malaysian company, has manufactured Deebag. It has made with polypropylene, woven mono and split filament Geotextile. Tencate Deebags are available in 1.2 m x 1.2 m x 1.2 or 1.5 m with a maximum safe working load of 3,000 kg. Due to the unavailability of Deebag, the research work was to be performed on a small scale. One square feet surface of the Deebag was collected from SNV Bangladesh, an international development organization situated in Khulna city. From the 0.1 $m^2$ or 1 $ft^2$ Deebag surface, 0.03 $m^3$ or 1 $ft^3$ wooden cistern was prepared, whose bottom portion was fitted with this Deebag surface. The top of the cistern was open and the sides were made of wood so that FS might be poured into the cistern at the top and pass through the Deebag surface at the bottom.

### Preparation of jute bags

Jute bags were prepared from burlap. This was chosen for dewatering as well as filtering purposes because it is locally available and jute has the potentiality of absorption capacity of heavy metal ions by jute fibred materials is assessed satisfactorily [15]. According to the design and study of Lee et al., prefabricated vertical drains (PVDs) were made from jute and two layers of jute burlap was used as filter sheath [16–18]. In the study of Bergado et al., One layer of jute geotextile filter sheath was used in making of PVDs [19]. By the literature review, three types of jute or burlap bags were hypothesized to be potentially as filtering media in the research work. These bags are single layer jute bag, double layer jute bag and two layer bag–one layer is jute and another is cotton cloth. Each bag was sized as 0.03 $m^3$ or 1 $ft^3$. Different types of burlaps are available in Khulna city. Among them whose pore size is 0.11 mm to 0.16 mm was selected for preparing the bags. The unit price of this type of jute per yard was BDT TK 72 or $0.85 or €0.72.

### Collection of polymer

A Polyacrylamide polymer was used in this study for dewatering. This polymer is commercially known as Aquaestrol 6370 polymer which is generally used for solid and liquid separation in municipal WW treatment. This polymer is marketed by Tianrun Chemical Company in China, of which Charisbrent Pte Ltd has technical, developmental and production collaborations. This polymer was collected with the help of Water & Sanitation for the Urban Poor (WSUP). The dosage of polymer is equal to 2 g per liter of FS solution.

## Methods

### Procedure of collection of samples

FS was collected from a three compartment septic tank which is situated in KUET campus. A bucket was used at first for collecting FS from the first compartment of the septic tank.

Because, first compartment of septic tank contains more sludge in compared with other two compartments. At the first time, no polymer was mixed with the FS. 35 kg equivalent FS was poured into every filtering or dewatering bag and from the FS, raw samples were also collected for laboratory analysis. The raw sample was not actually the supernatant but this is a homogeneous mix of FS liquor. Then for every bag, one sample was collected from the portion of FS passing by the filtration process through each bag for the laboratory assessment. At the same time, the portion of FS was retained inside the bags were weighted and recorded. The individual weight of every bag was also measured before. The dewatering as well as filtering process without using polymer were taken around 42 minutes by Deebag, 53 minutes by single layered jute bag and 61 minutes by double layered jute bag and jute and cotton layered bag.

FS was again collected in the same way described above from the same compartment of the septic tank. At the second time, polymer was used. Polymer was mixed with FS according to the mixing ratio and stirred for homogenous mixing for sometimes and finally, kept 1 hour. It was kept for 1 hour because, polymer takes the time to make flocs of FS. After that, the procedure was followed at the same as before. At that time with polymer, 16 minutes, 18 minutes for single layered jute bag and 22 minutes were required for the dewatering as well as filtering process by Deebag, single layered jute bag and both double layered jute bag and jute and cotton layered bag respectively.

## Procedure of laboratory analysis

The quality of the collected samples was analyzed with HACH 125 multimeter and HACHDR 2500 spectrophotometer in the laboratory according to Standard Methods for the examination of water and wastewater by American Public Health Association (APHA), 1998. The analyzed physical properties were biochemical oxygen demand ($BOD_5$), chemical oxygen demand (COD), pH, electrical conductivity (EC), chloride ($Cl^-$), phosphate ($PO_4^{3-}$), nitrate ($NO^{3-}$), total suspended solids (TSS) and total dissolved solid (TDS).

## Biochemical oxygen demand ($BOD_5$)

Dissolved oxygen (DO) of each sample were measured by HACH 125 multimeter. 5 mL of each sample was diluted with 295 mL of distilled water, i.e. dilution factor was 60. After measuring DO, all the samples were incubated for five days. The difference between DO at first day and DO at fifth day was multiplied by the dilution factor to find the $BOD_5$.

## Chemical oxygen demand (COD) and chloride ($Cl^-$)

Both COD and $Cl^-$ were measured by titration method. All samples are diluted with 100 dilution factor that was 1 mL of the sample was diluted with 99 mL of distilled water.

## pH and electrical conductivity (EC)

pH and EC were measured with the help of HACH 125 multimeter by inserting the probe of the multimeter into the samples directly. Stabled reading on the screen of the multimeter was taken.

## Phosphate ($PO_4^{3-}$) and nitrate ($NO^{3-}$)

For the assessment of phosphate and nitrate concentration in the samples, 50 dilution factor i.e. 1 mL of sample was diluted with 49 mL of distilled water. HACH DR 2500 spectrophotometer was used for the assessment. Phosver 3 and nitover 5 reagents were used for this

assessment respectively. All samples were prepared in two ways–one is 10 mL diluted sample and another is 10 mL reagent mixed diluted sample.

## Total suspended solid (TSS) and total dissolved solid (TDS)

TSS was measured by filtering with 1 μm pore size GF/C filters (Whatman). 100 mL of sample was taken. The difference between oven dried Whatman filter paper before and after filtration was the measurement of TSS per 100 mL and then it was converted into 1 mL. TDS was measured directly by HACH HQ14D portable meter.

## Results and discussions

### Treatment or filtration efficiency

**Biochemical oxygen demand (BOD$_5$).**   In Fig 1, high BOD$_5$ value was found both in raw WW and filtrated water for non-using polymer condition, but these values were found below the permissible limit set by the Department of Environment (DoE), Bangladesh for polymer using condition after filtration and BOD$_5$ of raw WW was also in low range. Since, raw sample was not any supernatant but collected from directly FS liquor, the high BOD$_5$ was observed in the raw sample. High BOD$_5$ in WW indicates that large concentration of organic matter presents in WW. Aerobic bacteria needs huge amount to oxygen to decompose the large concentration of organic matter and that causes a severe decrease of dissolved oxygen in the water. No aquatic life can survive below a certain amount of DO level [20]. From Fig 1, very low BOD$_5$ recovered was observed when no polymer was used. Deebag has found to remove maximum BOD$_5$ that was only 40%, whereas among jute bags, double layered bag removed 36.68%. Due to using polymer, the recovery percentage reached at 75.96% for Deebag and 65.38% for double layered jute bags. The higher recovery percentage was also observed in the case of the other two types of jute bags. BOD$_5$ concentration was dropped down to 62.4 mg/L that is 83.58% reduction of BOD$_5$ concentration from raw WW by only using polymer.

**Chemical oxygen demand (COD).**   Fig 2 shows a high COD value in raw WW for the non-using polymer condition because FS liquor was collected as raw sample. High COD is toxic and deleterious for the biological life as well as aquatic environment [21]. On the other

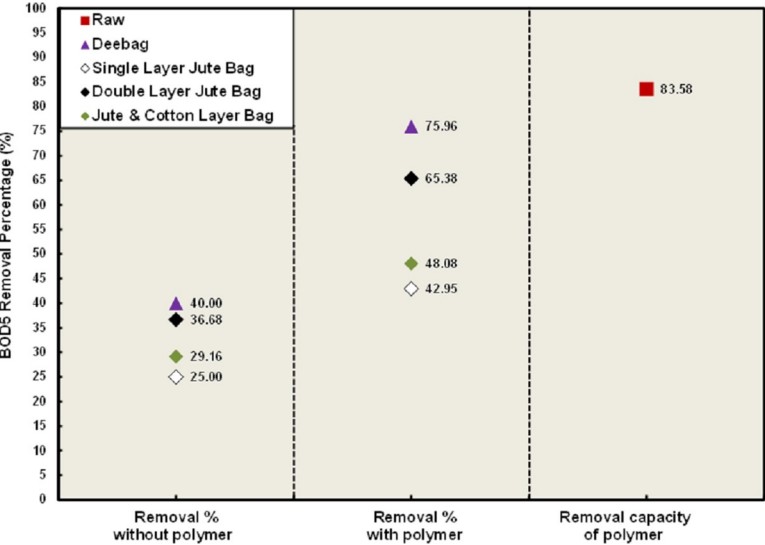

**Fig 1. Removal percentages of BOD$_5$.**

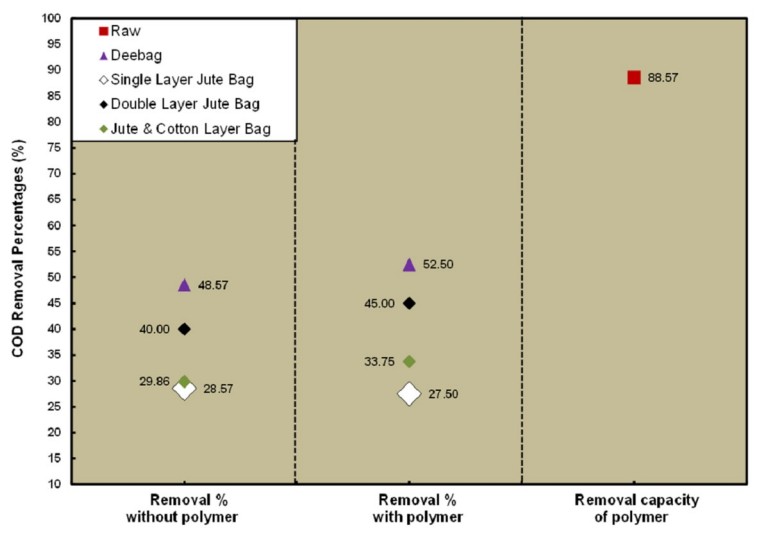

**Fig 2. Removal percentages of COD.**

hand, due to using polymer, low range COD value was found in the raw sample. Among the dewatering media, Deebag has shown best result in removing COD through filtration. It recovered 52.50% COD after filtration while using polymer and 48.57% without polymer. Doubled layered jute bag recovered maximum percentage of COD comparing other two types of jute bags. 45% COD was recovered by the double layered jute bag in polymer condition and 40% COD was in the non-polymer condition. Polymer by itself decreased 88.57% COD from raw WW. However, no filtering media is sufficiently capable to remove COD below the DoE permissible limit.

**pH.**  pH is the indication of acidic or alkaline condition of water. Table 1 shows the acidic condition of raw WW in non-using polymer condition. Filtrated samples were also found acidic. Both raw and filtrated samples without polymer was not within the DoE, Bangladesh Standard limit. Due to using polymer, pH value of raw WW was found as neutral that is within

**Table 1.  Concentration of different water quality parameter in different samples.**

| Parameter[a] | Without using polymer | | | | | With using polymer | | | | | DoE, BD Std. |
|---|---|---|---|---|---|---|---|---|---|---|---|
| | Raw water | Jute made bag | | | Deebag | Raw water | Jute made bag | | | Deebag | |
| | | SL | DL | JCL | | | SL | DL | JCL | | |
| $BOD_5$ | 380 | 285 | 240.6 | 269.2 | 228 | 62.4 | 35.6 | 21.6 | 32.4 | 15 | 40 |
| COD | 7000 | 5000 | 4200 | 4910 | 3600 | 600 | 420 | 355 | 400 | 300 | 200 |
| pH | 5.27 | 5.38 | 5.6 | 5.49 | 5.62 | 7.32 | 7.26 | 7.21 | 7.25 | 7.1 | 6.5–8.5 |
| EC | 3960 | 2750 | 2250 | 2660 | 2063 | 2800 | 2300 | 2030 | 2240 | 1640 | 1200 |
| $Cl^-$ | 8900 | 4330 | 4100 | 4250 | 3950 | 5880 | 3300 | 3040 | 3255 | 2875 | 600–1000 |
| $PO_4^{3-}$ | 118.3 | 97.15 | 91 | 95 | 82.25 | 51.6 | 44.6 | 38.4 | 43.7 | 23.3 | 35 |
| $NO_3^-$ | 430 | 315 | 301 | 315 | 280 | 240 | 165 | 140 | 156 | 135 | 250 |
| TSS | 4276 | 2590.32 | 1921.06 | 2464.2 | 1620 | 641 | 340.9 | 300.32 | 330 | 270.56 | 100 |
| TDS | 8851.28 | 3991.60 | 3549.98 | 3949.4 | 2712.86 | 1700.86 | 981.71 | 673.46 | 827.2 | 630 | 2100 |

[a]All values except pH and EC are expressed in mg/L and EC is expressed as μS/cm.

S = Single Layered; DL = Doubled Layer; JCL = Jute & Cotton Layered; DoE, BD Std. = Department of Environment, Bangladesh Standard.

the DoE, Bangladesh permissible limit. All filtrated water also became neutral in polymer using condition. Before and after using polymer, the pH levels of filtrate samples were in the same condition as that of the corresponding WW. So, pH level was not enhanced by the filtration process, but polymer improved the condition.

**Electrical conductivity (EC).** Electrical conductivity (EC) is usually a representation of salinity and Cl- is a major constitute of saline waters that directly affect the EC values [22,23]. EC value was comparatively higher than the standard because from the Table 1 higher amount of chloride content was found in all samples. Khulna is the south-west region of Bangladesh and these regions are highly affected by salinity intrusion. Since, KUET is situated in Khulna city and these areas are highly saline prone, that's why high electrical conductivity has been found in the raw wastewater samples. It has been found from Table 1 that the filtering media is not capable of removing electrical conductivity below the standard limit of DoE both in using and non-using polymer condition. It has been observed from Fig 3 that a good removing percentage was found both in the Deebag and the double layered jute bag. Removal percentage was near about the same for all filtering media in both conditions. However, the EC concentration in raw WW was decreased to 21.46% by the using of polymer.

**Chloride (Cl⁻).** Fig 4 shows a high chloride concentration in raw WW and filtrated samples. Khulna is a high saline prone area and chloride ions are the main constituents in saline water [24]. So, chloride concentration in water is normally high in this area. High level of chloride is toxic for aquatic plant which is not grown up in the saline zone [25].

Though chloride content in all samples with and without polymer was not below the permissible limit set by DoE, a good removal percentages was found by the filtering media. Deebag had the maximum removal capacity, whereas among jute bags, the maximum capacity was shown by the double layered jute bag. By using polymer, 33.93% chloride content was removed from the raw WW. From Fig 4, all filtering bags removed almost the same percentages in both conditions that indicates those bags were capable of removing a certain chloride content by the filtration process.

**Phosphate ($PO_4^{3-}$) and nitrate ($NO^{3-}$).** Municipal WWs are considered as the main sources of nutrient release, including phosphorous and nitrogen compounds in the aquatic environment as well as ecosystems because of using vast amounts of nutrients in human

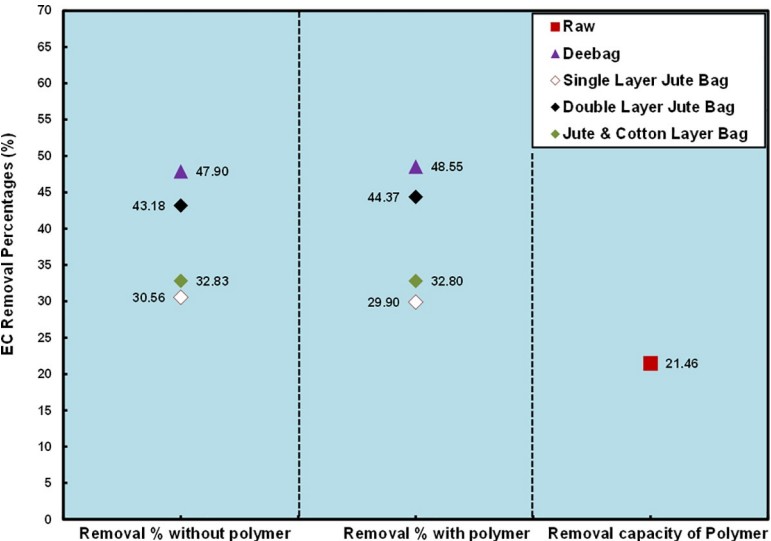

**Fig 3. Removal percentages of EC.**

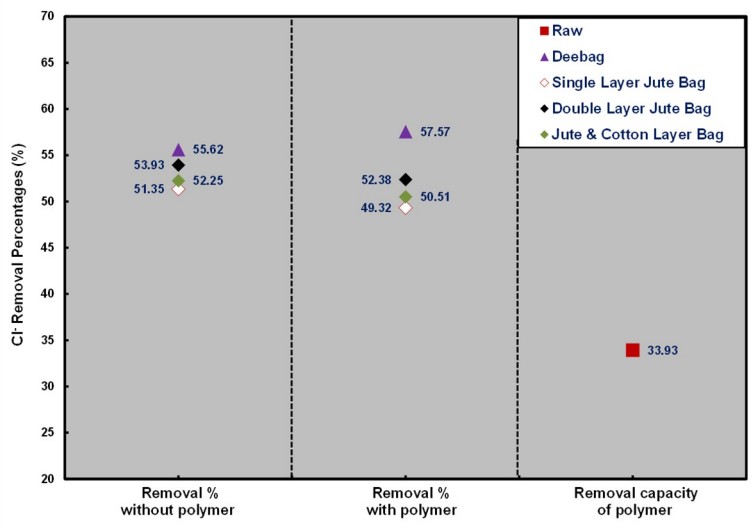

**Fig 4. Removal percentages of Cl⁻.**

being's daily life [26,27]. Discharging of nutrients such as phosphate and nitrate in water is a major concern because it promotes eutrophication that decrease the water quality by causing a rapid growth of aquatic plants such as algae [28–30] and results in the depletion of dissolved oxygen in the water $BOD_5$ that significantly influences the drinking water supplies and the fisheries economically [31,32].

The Low phosphate removal percentage was found by all filtering bags. Phosphate is above the DoE permissible limit. However, after filtering through Deebag and double layered jute bag, the phosphate concentration was found very close to the standard. Filtering bags were able to recover a certain amount of phosphate because, in Fig 5, each filter bag was found to recover almost same percentage of phosphate before and after using polymer. Maximum recovery was done by Deebag that was 30.47%, whereas only polymer reduced 56.38%

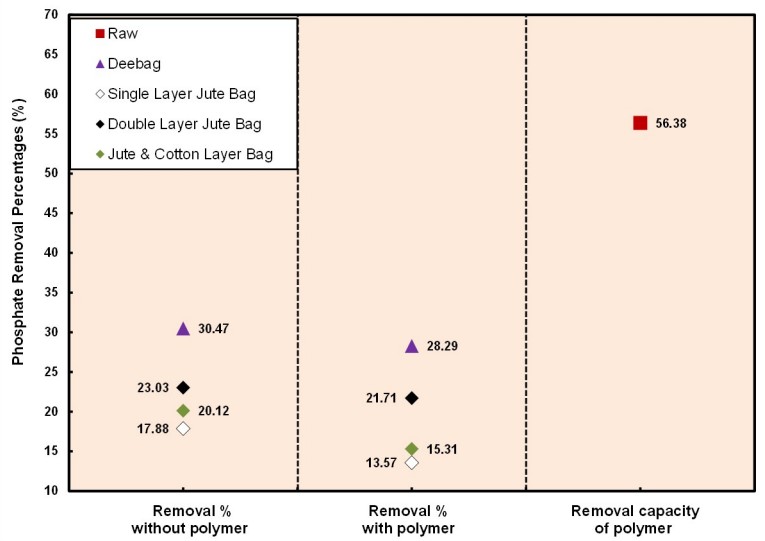

**Fig 5. Removal percentages of $PO_4^{3-}$.**

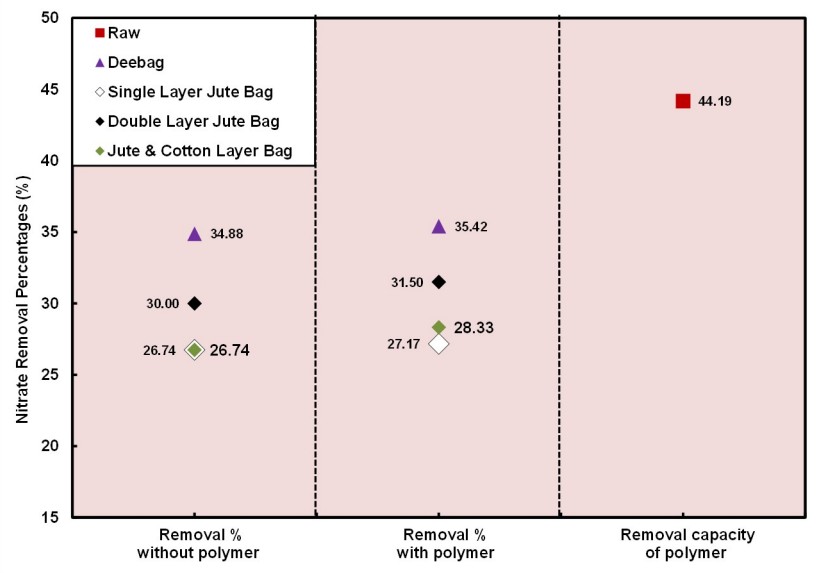

**Fig 6. Removal percentages of $NO_3^-$.**

phosphate content from WW. This removal percentage is not sufficient for jute bags. However, the output can be within the standard for low concentration of phosphate in WW.

Nitrate content is well below the permissible limit set by DoE for using polymer but that was vice-versa without polymer. It has been found in Fig 6 that 44.19% nitrate content was dropped down into the raw WW sample due to using polymer and after that it became below the permissible limit. Like phosphate, low removal percentage was also observed by the filtering bags. Before and after using polymer, the removal percentage was more or less stable for all filtering bags. So, all filtering bags are inefficient after reaching at a certain percent of removal of nitrate. The maximum removal percentage was found in the Deebag, whereas the double layered jute bag was better among other two jute filtering bags.

**Total suspended solid (TSS) and total dissolved solid (TDS).** TSS and TDS play an important role in the treatment of WW [33]. According to the study of Wu and Maskaly, a reduction in COD removal from WW was observed when TDS was more than 3000 mg/L [34]. The presence of TSS in water is the reason of depletion of oxygen level [33].

Table 1 shows a high amount of TSS and TDS not only in raw WW but also in the filtrated water sample without using polymer and were not met the permissible limit. Low concentration of TSS and TDS was found while using polymer. TDS in the filtrated water sample even in raw WW became below the standard limit set by DoE and TSS of all samples were very close to the permissible limit due to applying polymer. Around 80.78% TDS and 85.01% TSS concentration were recovered from the raw WW with using polymer. Though the Deebag showed the best performance, double layered jute bag also did better in comparison with other two type jute bags.

## Dewatering

According to Rose et al. and Strande et al., average 75% water is present in FS that means 25% is solid in an average [13,3]. Collivignarelli et. al. also mentioned in their study that biological sewage sludge typically contains 25 to 20% solids by weight [2]. So in this study, it was hypothesized that emptying of the FS pit by vacutug for safe disposal and cost minimization would be

**Table 2. Dewatering capacity of different bags.**

| Sample Name | Without mixing polymer | | | With mixing polymer | | |
|---|---|---|---|---|---|---|
| | Weight of Total FS (kg) | Retained Weight (kg) | Percentage of Retained (%) | Weight of Total FS (kg) | Retained Weight (kg) | Percentage of Retained (%) |
| Single jute bag | 35 | 2.985 | 8..53 | 35 | 6.416 | 18.33 |
| Double jute bag | | 3.895 | 11.13 | | 7.184 | 20.53 |
| Jute with cotton bag | | 3.332 | 9.53 | | 6.79 | 19.4 |
| Deebag | | 4.55 | 13 | | 7.841 | 22.4 |

possible due to the dewatering by the filtering bags. Table 2 shows the dewatering capacity of Deebag and three jute bags in both conditions of using polymer and not. Maximum retained or solid separation percentage was found 22.4% by the Deebag while using polymer and among three jute bags, the maximum percentage was found 20.53% that were done by the doubled layered jute bag. Without polymer, these percentages were reduced to approximately a half of those while using polymer for all filtering bags. So, considering 25% solid, maximum dewatering was found 88% for Deebag and among three type jute bags, 83% for double layered jute bags in polymer using condition.

## Conclusions

Dewatering and WW treatment are the most important concern in faecal sludge management. Higher the dewatering lower the WW treatment cost and complexities. The result of the study indicates that both dewatering and treatment efficiency by all the filtering bags were found better due to using polymer. Deebag has found in the best treatment performance and dewatering capacity in both conditions of using and non-using polymer, whereas the doubled layered jute bag is better than other two type of jute bags. 88% dewatering capacity has been shown by Deebag whereas it was 83% for double layered jute bags. Though $BOD_5$, pH and $NO_3^-$ were found below the standard limit set by DoE, Bangladesh after the treatment of WW by the filtration process through Deebag and three jute bags with using polymer, further treatment is required for COD, EC, $Cl^-$, $PO_4^{3-}$ and TSS parameters. All the filtering media or bags can be considered as effective in primary treatment and this will help to minimize the cost of the final treatment. But these filtering bags in polymer using condition have shown better performance for dewatering. Only using polymer is also considered as primary treatment of WW because only polymer has reduced a good percentage of concentration of the analyzed parameters in raw sample. Using polymer, 83.58% $BOD_5$ concentration has been reduced in raw WW and 88.57% COD was reduced. However, if the analyzed parameters present in low concentration in WW, the Deebag and the double layered jute bag are efficient to reduce the concentration of the parameters within the standard of DoE, Bangladesh.

## Supporting information

**S1 Table. Concentration and removal percentages of BOD5.**
(XLS)

**S2 Table. Concentration and removal percentages of COD.**
(XLS)

**S3 Table. Concentration and removal percentages of pH.**
(XLS)

**S4 Table. Concentration and removal percentages of electrical conductivity.**
(XLS)

**S5 Table. Concentration and removal percentages of chloride.**
(XLS)

**S6 Table. Concentration and removal percentages of phosphate.**
(XLS)

**S7 Table. Concentration and removal percentages of nitrate.**
(XLS)

**S8 Table. Concentration and removal percentages of TSS.**
(XLS)

**S9 Table. Concentration and removal percentages of TDS.**
(XLS)

## Acknowledgments

The authors would like to thank SNV, Bangladesh for arranging Deebag sample and also thank Water & Sanitation for the Urban Poor (WSUP) for supplying Aquaestrol 6370 Poly-acrylamide polymer.

## Author Contributions

**Conceptualization:** Muhammed Alamgir.

**Data curation:** Mehedi Hasan Mishuk.

**Formal analysis:** Mehedi Hasan Mishuk.

**Methodology:** Mehedi Hasan Mishuk.

**Project administration:** Muhammed Alamgir.

**Supervision:** S. M. Tariqul Islam, Muhammed Alamgir.

**Validation:** Muhammed Alamgir.

**Visualization:** Muhammed Alamgir.

**Writing – original draft:** Mehedi Hasan Mishuk.

**Writing – review & editing:** Mehedi Hasan Mishuk, S. M. Tariqul Islam, Muhammed Alamgir.

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
