## [Decision Letter · Decision Letter 0]

10 Jun 2020

PONE-D-20-12825

COMPARISON OF THE EFFICIENCY OF DEEBAG AND JUTE MADE BAG FOR FAECAL SLUDGE MANAGEMENT AND WASTEWATER TREATMENT

PLOS ONE

Dear Dr. Mishuk,

Thank you for submitting your manuscript to PLOS ONE. After careful consideration, we feel that it has merit but does not fully meet PLOS ONE’s publication criteria as it currently stands. Therefore, we invite you to submit a revised version of the manuscript that addresses the points raised during the review process.

We look forward to receiving your revised manuscript.

Kind regards,

Mentore Vaccari

Academic Editor

PLOS ONE

Journal Requirements:

4. Please ensure that you refer to Figures 3 and 6 in your text as, if accepted, production will need this reference to link the reader to the figure.

Reviewers' comments:

Reviewer's Responses to Questions

**Comments to the Author**

1. Is the manuscript technically sound, and do the data support the conclusions?

Reviewer #1: Partly

Reviewer #2: Partly

2. Has the statistical analysis been performed appropriately and rigorously? 

Reviewer #1: N/A

Reviewer #2: N/A

3. Have the authors made all data underlying the findings in their manuscript fully available?

Reviewer #1: Yes

Reviewer #2: Yes

4. Is the manuscript presented in an intelligible fashion and written in standard English?

Reviewer #1: Yes

Reviewer #2: No

5. Review Comments to the Author

Reviewer #1: The paper presents an interesting study on the comparison between the performance obtained with deebag and jute bags applied for the fecal sludge treatment.

The experimental work is quite complete, but several improvements are needed in order to clarify/integrate some experimental results. Moreover, I think that additional tests to confirm the results obtained could be very useful.

Thus, in my opinion, the paper needs significant revision before the publication.

My comments:

Abstract

1) Lines 20-21: I suggest to define the acronyms.

Materials and Methods

2) Line 123: I suggest to report the unit price also in Euro or US $.

3) Lines 130-131: I suggest to replace “Two grams of the polymer are required for every liter solution of FS” with “The dosage of polymer is equal to 2 g per liter of FS solution”.

4) Lines 141-142: The retained FS by the bags were weighted and recorded after the dewatering process. How many hours did the dewatering process ?

5) Line 153: I suggest to replace “(BOD)” with “(BOD5)”. Please, amend the acronym in the whole manuscript.

6) Line 154: Please, replace “electric conductivity” with “electrical conductivity”. Please, amend this term the whole manuscript.

7) Lines 157-195: I suggest to reduce these sections with minor details on methodology. Moreover, I think that this part can be include in 2.6, avoiding the numbering of subsections 2.6.1, 2.6.2, etc. Finally, in 2.6.4 Iron (Fe) is written, but any details on its measurement are reported.

Results and discussion

8) Line 201: BOD is a measure of the amount of oxygen required to remove waste organic matter from water in the process of decomposition by aerobic bacteria. Thus, BOD is often used as an index of the degree of organic pollution in wastewater. It is not a microbial condition of wastewater.

9) Table 1: I suggest to highlight (e.g. with bold fonts) the values higher than DoE BD Standard. Moreover, I think that EC is expressed as µS/cm.

10) Line 214: The sentence “BOD concentration was dropped down to 65.33% in raw WW by only using polymer” is wrong. BOD5 concentration was lower than 62.4 mg/L.

11) Figure 1: I suggest to remove the data referred to concentrations. These values are reported in Table 1. I think that a Figure with removal of pollutants is more suitable. The same comment can be apply to other Figures.

12) Section 3.1.4: I suggest to add comments on the pH values compared with the Bangladesh Standard.

13) Section 3.1.5: Could be an influence of polymer dosage on EC values?

Moreover, the sentence “However, the EC concentration in raw WW was decreased to 29.29% by the using of polymer” is wrong. The correct value is 21.46%.

Reviewer #2: I revised the paper titled COMPARISON OF THE EFFICIENCY OF DEEBAG AND JUTE MADE BAG FOR

FAECAL SLUDGE MANAGEMENT AND WASTEWATER TREATMENT for a possible publication in PLOS ONE. Results are interesting in particular for a possible application in a low-income country, but should be better and deepened commented. The language should be carefully improved. Several grammar mistakes are present in the manuscript. I suggest major revision.

My comments are the following:

1) Abstract should be revised and shortened focusing on main aspect of the research. Some sentences regarding the methods used are not necessary in the abstract (e.g. from line 18 to 22).

2) Line 33-35: In what sense “inappropriate management”. Please, describe better.

3) In the intro, also greywater (GW) should also be cited due to the fact that in low income countries they are often release into the environment without treatments and can contain several pollutants.

4) Line 62: “66,257”

5) Line 63: “her” should be amended with “its”. Please, revised all manuscript in order to amend this grammar mistakes.

6) Line 65, Please, describe “vacutags”.

7) Line 77-78: please rephrase

8) Line 78-79: “They are” not “It is”. Please, revise carefully all manuscript.

9) In the Introduction more recent literatures should be inserted. I suggest you some recent papers:

(sludge management and reuse)

Collivignarelli, M. C., Canato, M., Abba, A., Carnevale Miino, M. (2019). Biosolids: What are the different types of reuse?. Journal of Cleaner Production, 117844. https://doi.org/10.1016/j.jclepro.2019.117844

(greywater treatment with low impact processes)

Al-Gheethi, A. A. S., Noman, E. A., Mohamed, R. M. S. R., Bala, J. D., & Kassim, A. H. M. (2019). Qualitative Characterization of Household Greywater in Developing Countries: A Comprehensive Review. In Management of Greywater in Developing Countries (pp. 1-31). Springer, Cham. https://doi.org/10.1007/978-3-319-90269-2_1

Collivignarelli, M. C., Carnevale Miino, M., Gomez, F. H., Torretta, V., Rada, E. C., Sorlini, S. (2020). Horizontal Flow Constructed Wetland for Greywater Treatment and Reuse: An Experimental Case. International Journal of Environmental Research and Public Health, 17(7), 2317. https://doi.org/10.3390/ijerph17072317

10) At the end of the Introduction a paragraph with the main points that your research studied should be inserted.

11) In materials and methods, please divide materials and methods in two new subsection 2.1 and 2.2. It will be more clearer.

12) Line 96-98: Is this a result?

13) “a one cubic foot wooden cistern was prepared”. Please, use international units of measure.

14) Line 140-143. Please rephrase.

15) It is not necessary to describe in detailed the titration method to detect COD and Cl- It is well known.

16) Line 201. In what sense “BOD is a microbial condition of WW”. Please, described well.

17) Line 207: A high concentration of nitrate and COD not necessarily indicates presence of high concentration of BOD.

18) Line 207-209: Describe better.

19) Figures 1, 2, 3, 4, 5, and 6. What is BDS? Please, specify in the caption. Please, divide better the concentration and the removal percentages in the figures to make it clearer. Why Figure 1 is in COD section?

20) Similar figures could be merged into a Figure with more subfigures.

21) Please, review the conclusions and insert some percentages results.

6. PLOS authors have the option to publish the peer review history of their article (what does this mean?). If published, this will include your full peer review and any attached files.

Reviewer #1: No

Reviewer #2: No

---

## [Author Response · Author response to Decision Letter 0]

6 Sep 2020

Response to the Comments of Reviewer#1

Abstract

1) Lines 20-21: I suggest to define the acronyms.

Response: I have defined the acronyms in line 23-26 of the manuscript with track changes.

Materials and Methods

2) Line 123: I suggest to report the unit price also in Euro or US $.

Response: I have included the unit price both in Euro and US $ in line 144-145of the manuscript with track changes.

3) Lines 130-131: I suggest to replace “Two grams of the polymer are required for every liter solution of FS” with “The dosage of polymer is equal to 2 g per liter of FS solution”.

Response: The line has been replaced in line 153-154of the manuscript with track changesas per the suggestion.

4) Lines 141-142: The retained FS by the bags were weighted and recorded after the dewatering process. How many hours did the dewatering process?

Response: New lines (183-186 and 190-193) have been added in the manuscript with track changes regarding the time taken by dewatering process.

5) Line 153: I suggest to replace “(BOD)” with “(BOD5)”. Please, amend the acronym in the whole manuscript.

Response: BOD has been replaced by BOD5 in the whole manuscript.

6) Line 154: Please, replace “electric conductivity” with “electrical conductivity”. Please, amend this term the whole manuscript.

Response: Electric Conductivity has been replaced by Electrical Conductivity in the whole manuscript.

7) Lines 157-195: I suggest to reduce these sections with minor details on methodology. Moreover, I think that this part can be include in 2.6, avoiding the numbering of subsections 2.6.1, 2.6.2, etc. Finally, in 2.6.4 Iron (Fe) is written, but any details on its measurement are reported.

Response: “2.6.4 iron (Fe)” was a typing mistake. I have divided the materials and methods into two sections– one is Materials and other is Methods. I have removed all subsections. This was also recommended from reviewer 2.

Results and discussion

8) Line 201: BOD is a measure of the amount of oxygen required to remove waste organic matter from water in the process of decomposition by aerobic bacteria. Thus, BOD is often used as an index of the degree of organic pollution in wastewater. It is not a microbial condition of wastewater.

Response: Yes, I agree. It’s my fault. I have omitted this sentence.

9) Table 1: I suggest to highlight (e.g. with bold fonts) the values higher than DoE BD Standard. Moreover, I think that EC is expressed as µS/cm.

Response: All the values which was higher than the DoE, BD standard have been highlighted with bold fonts in the manuscript. EC has been also expressed as µS/cm.

10) Line 214: The sentence “BOD concentration was dropped down to 65.33% in raw WW by only using polymer” is wrong. BOD5 concentration was lower than 62.4 mg/L.

Response: This was typing mistake. I have made correction and now, the sentence is in the line 270-271 of the manuscript with track changes.

11) Figure 1: I suggest to remove the data referred to concentrations. These values are reported in Table 1. I think that a Figure with removal of pollutants is more suitable. The same comment can be apply to other Figures.

Response: Since, the concentrations of different parameter were mentioned in table 1, so all existing figures have been replaced by the figures which have showed only removal percentage (applied for figure 1 to 6).

12) Section 3.1.4: I suggest to add comments on the pH values compared with the Bangladesh Standard.

Response: A comment on the pH values compared with Bangladesh Standard has been added in the line 296-298 of the manuscript with track changes.

13) Section 3.1.5: Could be an influence of polymer dosage on EC values?

Moreover, the sentence “However, the EC concentration in raw WW was decreased to 29.29% by the using of polymer” is wrong. The correct value is 21.46%.

Response: Yes, I made typing mistake and I have already corrected the value in the line 320 of the manuscript with track changes. Since, due to applying polymer, EC values were decreased in all samples, so it could be an influence of polymer dosage. It needs another study for this issue.

Response to the Comments of Reviewer #2

1) Abstract should be revised and shortened focusing on main aspect of the research. Some sentences regarding the methods used are not necessary in the abstract (e.g. from line 18 to 22).

Response: Abstract has been revised and shortened as per your suggestions. 

2) Line 33-35: In what sense “inappropriate management”. Please, describe better.

Response: Inappropriate management was sensed to mean “lack of proper management”. FSM is a combination of the process of collection, transportation, deposition, treatment and end-use. Proper management indicates the right way to collect, transport, deposit, treat and end-use. “Inappropriate management” has been replaced by “lack of proper management”. 

3) In the intro, also greywater (GW) should also be cited due to the fact that in low income countries they are often release into the environment without treatments and can contain several pollutants.

Response: Yes, it is right. Here, I have actually focused on only faecal sludge. Because, the raw samples were collected from the septic of Khulna University of Engineering and Technology (KUET). And there, greywater is directly discharged into public storm drain. Not only in the university but also it is a common scenario in the city areas of Bangladesh. So, only faecal matters are come to the septic tank and that’s why greywater is not considered in this study. Introduction has been also revised (e.g. line 59-65 of the manuscript with the track changes)

4) Line 62: “66,257”

Response: I have corrected as “66,257 number of households” in the line 73of the manuscript with track changes.

5) Line 63: “her” should be amended with “its”. Please, revised all manuscript in order to amend this grammar mistakes.

Response: It has been amended as “its” in the line 74 of the manuscript with track changes.

6) Line 65, Please, describe “vacutags”.

Response: Vacutags are described in the line 78-80 of the manuscript with track changes.

7) Line 77-78: please rephrase

Response: Those lines have been rephrased in the manuscript which are found as line 89-93 with track changes.

8) Line 78-79: “They are” not “It is”. Please, revise carefully all manuscript.

Response: “It is” has been replaced by “They are” in the line 92of the manuscript with track changes.

9) In the Introduction more recent literatures should be inserted. I suggest you some recent papers:

(sludge management and reuse) Collivignarelli, M. C., Canato, M., Abba, A., CarnevaleMiino, M. (2019). Biosolids: What are the different types of reuse?. Journal of Cleaner Production, 117844. https://doi.org/10.1016/j.jclepro.2019.117844

(greywater treatment with low impact processes) Al-Gheethi, A. A. S., Noman, E. A., Mohamed, R. M. S. R., Bala, J. D., &Kassim, A. H. M. (2019). Qualitative Characterization of Household Greywater in Developing Countries: A Comprehensive Review. In Management of Greywater in Developing Countries (pp. 1-31). Springer, Cham. https://doi.org/10.1007/978-3-319-90269-2_1

*-Collivignarelli, M. C., CarnevaleMiino, M., Gomez, F. H., Torretta, V., Rada, E. C., Sorlini, S. (2020). Horizontal Flow Constructed Wetland for Greywater Treatment and Reuse: An Experimental Case. International Journal of Environmental Research and Public Health, 17(7), 2317. https://doi.org/10.3390/ijerph17072317

Response: Since, greywater has not been considered in this study due to the reason mentioned in the Response of comment#3, so nothing is required to cite from the 2nd and 3rd reference mentioned in the comment above. However, from the first reference, it has been cited in line 41-43 and 384-385 of the manuscript with track changes.

10) At the end of the Introduction a paragraph with the main points that your research studied should be inserted.

Response: At the last paragraph of Introduction, the main points of the research has been inserted in the line 96-98 of the manuscript with track changes.

11) In materials and methods, please divide materials and methods in two new subsection 2.1 and 2.2. It will be more clearer.

Response: Materials and methods has been divided in two subsection – one is materials and another is method.

12) Line 96-98: Is this a result?

Response: It can be considered as result. From the practical experience and talking to the local people, it has found that transportation cost of FS from the on-site sanitation system to the KCC designated disposal site was higher from other parts of the Khulna City than the study area. Because, the study area is located at the most away from the designated disposal site. This findings is not the scope of the study. Moreover, the study area was selected based on this key point.

13) “a one cubic foot wooden cistern was prepared”. Please, use international units of measure.

Response: one cubic foot has been converted into SI unit and this has been incorporated in the line 124-125 of the manuscript with track changes.

14) Line 140-143. Please rephrase.

Response: Those lines have been rephrased in the manuscript which are found as line 181-184 of the manuscript with track changes.

15) It is not necessary to describe in detailed the titration method to detect COD and Cl- It is well known.

Response: Ok. I have omitted the details of titration method in the manuscript.

16) Line 201. In what sense “BOD is a microbial condition of WW”. Please, described well.

Response: Actually, BOD is not fully microbial condition. It’s my fault to write. I have omitted this sentence. Rather, BOD is a measure of the amount of oxygen required to remove waste organic matter from water in the process of decomposition by aerobic bacteria

17) Line 207: A high concentration of nitrate and COD not necessarily indicates presence of high concentration of BOD.

Response: Yes, it is right. I have deleted this line from the manuscript to avoid controversy. 

18) Line 207-209: Describe better.

Response: Those lines have been described better in the manuscript which are found as line 260-263 with track changes.

19) Figures 1, 2, 3, 4, 5, and 6. What is BDS? Please, specify in the caption. Please, divide better the concentration and the removal percentages in the figures to make it clearer. Why Figure 1 is in COD section?

Response: Concentration portion has been removed from all the figures and only removal percentages have been shown in those figures. This has been made to avoid repetition because table 1 has already showed the concentration of different parameters. Since, concentration portion of the graph has been removed, so BDS has been also removed. Only removal of percentages have been shown in all the graphs.

I am sorry that it was formatting mistake and figure 1 has been placed in the BOD5 section.

20) Similar figures could be merged into a Figure with more subfigures.

Response: It may not be required because Concentration portion has been removed from all the figures and only removal percentages have been shown in those figures.

21) Please, review the conclusions and insert some percentages results.

Response: Conclusions have been reviewed and some percentage results have been included in the line 410-412 and 420-424 of the conclusion section of the manuscript with track changes.

---

## [Decision Letter · Decision Letter 1]

22 Sep 2020

PONE-D-20-12825R1

COMPARISON OF THE EFFICIENCY OF DEEBAG AND JUTE MADE BAG FOR FAECAL SLUDGE MANAGEMENT AND WASTEWATER TREATMENT

PLOS ONE

Dear Dr. Mishuk,

Thank you for submitting your manuscript to PLOS ONE. After careful consideration, we feel that it has merit but does not fully meet PLOS ONE’s publication criteria as it currently stands. Therefore, we invite you to submit a revised version of the manuscript that addresses the points raised during the review process.

We look forward to receiving your revised manuscript.

Kind regards,

Mentore Vaccari

Academic Editor

PLOS ONE

Reviewers' comments:

Reviewer's Responses to Questions

**Comments to the Author**

1. If the authors have adequately addressed your comments raised in a previous round of review and you feel that this manuscript is now acceptable for publication, you may indicate that here to bypass the “Comments to the Author” section, enter your conflict of interest statement in the “Confidential to Editor” section, and submit your "Accept" recommendation.

Reviewer #1: (No Response)

Reviewer #2: (No Response)

2. Is the manuscript technically sound, and do the data support the conclusions?

Reviewer #1: Yes

Reviewer #2: (No Response)

3. Has the statistical analysis been performed appropriately and rigorously? 

Reviewer #1: N/A

Reviewer #2: (No Response)

4. Have the authors made all data underlying the findings in their manuscript fully available?

Reviewer #1: Yes

Reviewer #2: (No Response)

5. Is the manuscript presented in an intelligible fashion and written in standard English?

Reviewer #1: Yes

Reviewer #2: (No Response)

6. Review Comments to the Author

Reviewer #1: The authors have amended the manuscript according to my comments. In my opinion, the paper could be published after a minor revision.

My comments:

1) Please, use the same word for vacutug. In the lines 77, 89 and 91 (related to clean version) “vacutug” is used; in the lines 83 and 85 “vacutag” is adopted.

2) Replace PO43+ with PO43- (e.g. in the line 192 of clean version). Please, check the whole manuscript. Moreover, I suggest to write BOD5 with “5” in subscript mode.

3) In the manuscript with track of change mode, the Figure 3 does not show the removal percentage of EC. In the clean version the Figure 3 is corrected.

4) Line 358 (of clean version): I suggest to replace “According to Rose and Strande et al.,” with “According to Rose et al. and Strande et al.”.

5) Line 385 (of clean version): “... for other analyzed parameters”. I suggest to specify the other parameters.

Reviewer #2: I revised the paper and I found that my suggestions, where possible, have been implemented. Therefore, in my opinion this paper can be published on this journal.

7. PLOS authors have the option to publish the peer review history of their article (what does this mean?). If published, this will include your full peer review and any attached files.

Reviewer #1: No

Reviewer #2: No

---

## [Author Response · Author response to Decision Letter 1]

26 Sep 2020

Response to the Comments of Reviewer#1

1) Please, use the same word for vacutug. In the lines 77, 89 and 91 (related to clean version) “vacutug” is used; in the lines 83 and 85 “vacutag” is adopted.

Response: “Vacutug” is the right word. I have replaced vacutag with vacutug in the lines 83 and 85 of the clean version. It has been also corrected in the manuscript with track changes.

2) Replace PO43+ with PO43- (e.g. in the line 192 of clean version). Please, check the whole manuscript. Moreover, I suggest to write BOD5 with “5” in subscript mode.

Response: I have replaced PO43+ with PO43- in the clean version of manuscript. I have also write BOD5 with BOD5 that is “5” in subscript mode in the clean version of the manuscript. Both changes have also been done in the manuscript with track changes.

3) In the manuscript with track of change mode, the Figure 3 does not show the removal percentage of EC. In the clean version the Figure 3 is corrected.

Response: This was my mistake and I am extremely sorry for that. I have made change the Figure 3 with the removal percentage of EC in the manuscript with track changes.

4) Line 358 (of clean version): I suggest to replace “According to Rose and Strande et al.,” with “According to Rose et al. and Strande et al.”.

Response: In the line 358 of clean version of the manuscript, “According to Rose and Strande et al.,” has been replaced with “According to Rose et al. and Strande et al.”. This change has also been done in the manuscript with track changes.

5) Line 385 (of clean version): “... for other analyzed parameters”. I suggest to specify the other parameters.

Response: The other analyzed parameters have been specified as COD, EC, Cl-, PO43- and TSS in the line 385 of clean version of the manuscript. Same changes have been also done in the manuscript with track changes.

---

## [Editor Report · Decision Letter 2]

8 Oct 2020

COMPARISON OF THE EFFICIENCY OF DEEBAG AND JUTE MADE BAG FOR FAECAL SLUDGE MANAGEMENT AND WASTEWATER TREATMENT

PONE-D-20-12825R2

Dear Dr. Mishuk,

We’re pleased to inform you that your manuscript has been judged scientifically suitable for publication and will be formally accepted for publication once it meets all outstanding technical requirements.

Kind regards,

Mentore Vaccari

Academic Editor

PLOS ONE

---

## [Editor Report · Acceptance letter]

12 Oct 2020

PONE-D-20-12825R2 

Comparison of the efficiency of Deebag and Jute made bag for faecal sludge management and wastewater treatment 

Dear Dr. Mishuk:

I'm pleased to inform you that your manuscript has been deemed suitable for publication in PLOS ONE. Congratulations! Your manuscript is now with our production department. 

Kind regards, 

on behalf of

Professor Mentore Vaccari 

Academic Editor

PLOS ONE